# Crafting the Future: Machine Learning for Ocean Forecasting

Patrick Heimbach[1], Fearghal O'Donncha[2], Timothy A. Smith[3], Jose Maria Garcia-Valdecasas[4], Alain Arnaud[5], Liying Wan[6]

[1]Oden Institute for Computational Engineering and Sciences, The University of Texas at Austin, Austin, TX, United States
[2]IBM Research, Dublin, Ireland
[3]NOAA Physical Sciences Laboratory, Boulder, CO, United States
[4]Nologin Oceanic Weather Systems, Santiago de Compostela, Spain
[5]Mercator Ocean International, Toulouse, France
[6]National Marine Environmental Forecasting Center Beijing, China

*Correspondence to*: Patrick Heimbach (heimbach@utexas.edu)

**Abstract.** Artificial intelligence and machine learning are accelerating research in Earth system science, with huge potential for impact and challenges in ocean prediction. Such algorithms are being deployed on different aspects of the forecasting workflow with the aim of improving its speed and skill. They include pattern classification and anomaly detection, regression and diagnostics, state prediction from nowcasting to synoptic, sub-seasonal, and seasonal forecasting. This brief review emphasizes scientific machine learning methods that have the capacity to embed domain knowledge, to ensure interpretability through causal explanation, to be robust and reliable, to involve effectively high dimensional statistical methods, supporting multi-scale and multi-physics simulations aimed at improving parameterization, and to drive intelligent automation as well as decision support. An overview of recent numerical developments is discussed, highlighting the importance of fully data-driven ocean models for future expansion of ocean forecasting capabilities.

## 1 Introduction

Research into applications of artificial intelligence (AI) and machine learning (ML) in ocean, atmospheric and climate sciences has accelerated at a breathtaking pace over the last 5 years or so (e.g., Schneider et al., 2023; Eyring et al., 2024). With essentially all these applications concerned with ML, we will drop the more broadly defined "AI" term in most of the following, except when used by references cited. We will also take the perspective of scientific machine learning (SciML), defined in a 2019 U.S. Department of Energy report on "Basic Research Needs for Scientific Machine Learning" (Baker et al., 2019), which emphasizes six key elements of SciML algorithms: (i) ML approaches that incorporate domain knowledge, such as physical principles, symmetries, constraints, expert feedback, computational simulations, and formal uncertainties; (ii) ML approaches that are interpretable, such that user's confidence in ML-based model predictions may be bolstered by causal explanations based on a user's domain knowledge; (iii) ML approaches that are robust and reliable as a prerequisite for making high-stakes, high-regret decisions; (iv) ML approaches that are data-intensive, i.e., that ingest high-dimensional, noisy, and uncertain input data which contain complex structures and which require statistical and probabilistic methods to deal with ill-

conditioning, non-uniqueness, and over-fitting; (v) ML approaches that enhance modeling and simulation to support, e.g., multi-scale, multi-physics simulations in terms of improved model parameterization or model acceleration; and (vi) ML approaches to support intelligent automation and decision support, which can range from quality control to application-oriented post-processing workflows. Arguably, all of these criteria are fundamental to the uses of ML in ocean prediction.

Next, following the review by Reichstein et al. (2019), it is useful to distinguish different categories of ML applications, (A) classification and anomaly detection, which is concerned with, e.g., finding extreme event patterns or the classification of important structures or regimes; (B) regression, which is concerned with state reconstruction of important state variables, parameters, or diagnostics (metrics) from available data; and (C) state prediction, ranging from nowcasting to operational forecasting, to sub-seasonal to seasonal prediction. A comprehensive collection of review articles on deep learning in Earth sciences is Camps-Valls et al. (2021), covering both algorithmic foundations as well as examples of all three categories.

Because the subject of this document is on ocean prediction, we will focus the following on the third category, state prediction or forecasting. To keep this review manageable, we will not review the interesting subjects of ML applications for state reconstruction, downscaling, or classification.

## 2 State prediction

The workflow of operational ocean prediction largely follows that of numerical weather prediction (NWP). Its core engine is a data assimilation (DA) framework, consisting of a physical model, i.e., a complex algorithm for solving a set of partial differential equations (PDEs), a workflow for quality-controlling and ingesting diverse observational data streams into the DA system (ideally in near-real time), and an optimal estimation algorithm that combines models and data in a formal manner that produces statistically optimal forecasts (e.g., Park and Zupanski, 2022). As pointed out by S. Penny in a 2022 U.S. National Academy of Sciences workshop on Machine Learning and Artificial Intelligence to Advance Earth System Science (NASEM, 2022), ML approaches hold the prospect for accelerating various elements of the DA workflow. We briefly summarize ML approaches targeting the physical model as well as the DA algorithm. Opportunities in the application of ML for partial differential equation (PDE) based models fall into two main categories, one concerned with targeted insertion of ML within a physical model, the other with the complete replacement of the physical model by a surrogate model. In the former, certain elements or subcomponents of a physical model are replaced by a surrogate model (e.g., a neural network), whereas in the latter, the entire model is emulated. Chantry et al. (2021) have used the terms "soft AI" versus "hard AI". We avoid the somewhat non-descriptive or ambiguous terminology in order not to give a false sense of which of these approaches is "harder" to realize.

### 2.1 Hybrid physics-ML models: enhancing forecast models and data assimilation with ML algorithms

A major source of model uncertainty is the parameterization of subgrid-scale (SGS) processes, both in terms of structural errors (formulation of functional representations of parameterizations) as well as parametric uncertainties (calibrating empirical

parameters in the functional representations). Exciting efforts are underway to apply machine learning to replace conventional functional representations subgrid-scale (SGS) turbulent oceanic processes with surrogate models that are based on machine learning and that have been learned either offline or online (Bolton and Zanna, 2019; Frezat et al., 2021a, 2021b; Zhang et al., 2023; Sane et al., 2023; Perezhogin et al., 2023b). This follows on early ideas in the context of climate model parameterization

(e.g., Schneider et al., 2017; Rasp et al. 2018). Similarly, equation discovery has proven successful to infer the functional form of such SGS ocean parameterization schemes (Zanna and Bolton, 2020, 2021; Perezhogin et al., 2023a). A longer list of related efforts exists for numerical weather prediction and has been reviewed by Dueben et al. (2021) and Boualègue et al. (2024). These surrogates, mostly some form of neural networks, have been trained on (i.e., fit to) what are considered simulations of much higher fidelity where these processes are resolved (e.g., large eddy simulations). Related efforts aim at learning improved

parameterizations from online bias correction or analysis increments incurred in sequential data assimilation (e.g., Gregory et al., 2023, 2024; Storto et al., 2024). Rapid progress is expected on this front in the coming years.

A second important application of hybrid approaches is the desire to replace specific numerical algorithms within PDE-based models by surrogate models to accelerate the simulation's time-to-solution. Studies exist within the generic field of computational fluid dynamics (Kochkov et al. (2021) and atmospheric modelling (Arcomano et al., 2023; Kochkov et al.,

2024), and with ocean-specific applications currently underway. Most of these take advantage of the concept of differentiable programming (Gelbrecht et al., 2023; Shen et al., 2023; Zhang et al., 2023; Sapienza et al., 2024). The underlying idea is to be able to generate code for the derivative of the physical model, in particular the adjoint model that enables efficient "online" (or "full model") learning of model parameters (or neural network weights).

There is a strong conceptual correspondence between machine learning and data assimilation (e.g., Abarbanel et al., 2018).

This provides various opportunities for embedding ML approaches within operational data assimilation workflows deployed in ocean prediction. Examples in ocean modeling so far are largely restricted to "toy problems" (such as the "Lorenz 96 model") or reduced-order versions of Earth system models but targeting eventual applications for ocean prediction (Bocquet et al., 2020; Brajard et al., 2021; Penny et al., 2022). The use of hybrid DA/ML approaches, be it in the context of ensemble DA or adjoint-based methods (e.g., 4DVar) presents substantial algorithmic hurdles (e.g., availability of a differentiable

dynamical core in the context of adjoint-based DA), which explains the relative paucity of such studies to date compared to purely data-driven methods.

## 2.2 Purely data-driven models: replacing numerical simulations with surrogate models

Over the last decade, with the acceleration of AI based solutions in other fields, a number of approaches to model the atmosphere and ocean using different hard AI have been developed. The overwhelming majority of these cases have so far

been realized in weather prediction or computational fluid dynamics.

### 2.2.1 Deterministic applications in weather prediction

Arguably, the field of data-driven weather forecasting has seen the strongest advances over the last five years, or so. This is a strong incentive for providing a very brief review organized in terms of approaches as a function of underlying "blocks" of

ML architectures employed. In a number of cases these architectural blocks are being combined. For example, the European Centre for Medium-Range Weather Forecast's AIFS (Lang et al., 2024) uses an overall "encode-process-decode" architecture, with a graph-based encoder and decoder, but a sliding window transformer as the processor.

***Convolutional Neural Networks (CNNs):*** Perhaps among the first serious endeavours using ML for emulating weather forecast models have been the uses of CNNs by Weyn et al., 2019; Weyn et al., 2020; Weyn et al., 2021; Karlbauer et al., 2023. CNNs use a mathematical operation called convolution to compress information, learning features or patterns in the input. Most recently, CNNs have been used by Cresswell-Clay et al., (2024) to create a coupled atmosphere-ocean emulator which produces a stable climate for 1,000-year periods and appears to be competitive with many CMIP6 models.

***Graph Neural Networks:*** Among the leading emulators for medium-range weather forecasts is the work by Lam et al. (2023). Based on graph neural networks, the GraphCast model was trained on atmospheric reanalysis data to produce autoregressive forecasts for up to 10 days.

***Transformers:*** These have been revolutionary in other ML/AI fields, such as natural language processing and image recognition/generation. They serve as the backbone for some of the leading atmospheric emulators, including Pangu Weather (Bi et al., 2023), FuXi (L. Chen et al., 2023), and FengWu (K. Chen et al., 2023).

***Fourier Neural Operators (FNOs):*** FNOs have been designed to move toward mesh-independent operators using Fourier bases (Li et al. 2020). FourCastNet (Pathak et al., 2022, Kurth et al. 2023) is based on a variant, the Adaptive FNO (AFNO). Another variant, the Spherical FNO (SFNO, Bonev et al. 2023; Watt-Meyer et al., 2023) seeks to take advantage of the spherical geometry (and underlying symmetries) in representing operator kernels for global-scale applications. Very recently, the use of SFNOs has been extended to coupled atmosphere-ocean modeling targeting seasonal prediction (C. Wang et al., 2024).

***Recurrent Neural Networks (including Long Short-Term Memory LSTM and Reservoir Computing):*** Recurrent neural networks (RNNs) are well-suited for sequential data processing, such as time series. Among special cases of RNNs, Long Short-term Memory (LSTM) networks use a special type of neuron that keeps track of previous inputs (short-term memory) and are especially useful for predicting time-series with memory, such as is the case for the atmosphere and ocean. Reservoir Computing (RC), another method based on RNNs with a pool of interconnected neurons forming the "reservoir", is particularly well adapted to the emulation of time series (e.g., Arcomano et al., 2020, Penny et al., 2022, Platt et al., 2023, Smith et al., 2023).

### 2.2.2 Probabilistic approaches – generative models

Most examples sketched in Section 2.2.1 describe emulators that are trained to be deterministic forecast models. Recent developments in ML have considered generative frameworks, i.e., models that are designed to be probabilistic. Such frameworks would include Variational Auto Encoders, Generative Adversarial Networks (GANs), and Diffusion Models. However, we note that GANs can suffer from a lack of sample diversity (Bayat 2023) and they are notoriously challenging to train, requiring careful setup to avoid training instabilities (e.g., Miyato et al., 2018). Moreover, in recent years Diffusion Models have started to outperform GANs in image classification (Dhariwal and Nichol, 2021). For these reasons, Diffusion

Models have become popular in generative modeling, despite their relatively high computational cost. Recent examples of Diffusion Models include GenCast (Price et al., 2024). Finally, we note a very recently developed technique DYffusion (Cachay et al., 2023; 2024), which is a generative framework that aims to reduce the computational cost of Diffusion Modeling by encoding the temporal evolution expected in physical systems into the generative process.

### 2.2.3 Physics-informed machine learning

The results of purely data-driven solutions may potentially produce meaningless output, as the training strategy of a neural network is to minimize a mathematical loss function, e.g., the mean squared error (i.e., L2 norm) between the prediction and the original target. Similar issues, e.g., producing overly blurred output, may arise with other choices of the loss function, such as an L1 norm. An evolution of this approach is to include some physical constraints in the loss function in order to force the ML algorithm to produce more consistent outputs, as the Navier-Stokes equation (Ma et al., 2022; Daw et al., 2021). This class of methods is known as physical-informed neural networks (PINNs). However, the performance of PINNs for extrapolation remains subject to debate (e.g., Du et al., 2023 for a cautionary example). Recently, another approach, which tries to solve differential equations using neural networks, is under development. Although this method is mostly developed for other physics fields, the methodology and knowledge can be applied to ocean modeling (Zubov et al., 2021; Smets et al., 2023).

### 2.2.4 Applications in ocean surface state forecasting

With previous examples mostly limited to weather prediction and computational fluid dynamics (in a few cases), we turn our attention to applications in the context of predicting ocean surface properties. They include the use of multi-layer perceptrons (James et al., 2018, Gracia et al., 2021) and LSTMs (Minuzzi and Farina, 2023; Lawal et al, 2024) for surface wave prediction, surface wave-current interaction forecasting, storm surge forecasting (Xie et al., 2023) and sea surface temperature prediction via deep learning (Wolff et al., 2020; Xu et al., 2023); the use of neural networks for accelerating resonant nonlinear wave-wave interaction in an ocean surface wave model (Puscasu, 2014), for regional to coastal sea level prediction (Nieves et al., 2021), for ocean color mapping (S. Chen et al., 2019), and for statistical downscaling (Accarino et al., 2021). Other applications include estimating ocean surface circulation (Sinha and Abernathey, 2021; Subel and Zanna, 2024) and predicting dissolved oxygen across scales (O'Donncha et al., 2022).

### 2.3 ML-based Ocean Circulation Prediction

Among the challenges of fully realizing the opportunities of ML approaches in ocean circulation prediction is the fact that, in the absence of adequate, densely sampled observational data, most ML applications rely on the use of data obtained from high-fidelity model simulations as training data sets. These data sets are very expensive to generate, limited in the temporal ranges that they can represent, remain subject to unquantified structural and parametric model uncertainty, require vast amounts of storage (order of Petabytes), and are thus challenging to query. Cloud-based solutions are the most promising approach for ubiquitous data access and analysis capabilities "close to the data" (Abernathey et al., 2020).

Within the realm of machine learning (ML) applications for ocean forecasting, progress has been somewhat limited. Recent developments have marked a shift in this landscape, particularly with the introduction of Fourier Neural Operators for modeling

oceanic processes, as suggested by Bire et al. (2023), Chattopadhyay et al. (2023), and Sun et al. (2024). These studies present fully data-driven ocean models that match the capabilities of traditional numerical ocean models in predicting high-resolution sea surface height (SSH) fields. FNOs are attractive for their performance in learning complex, high-dimensional mappings and their ability to incorporate physical laws and constraints, which are prominently observable in the spectral domain. A

drawback of FNOs applied to ocean (unlike atmospheric) modelling is the existence of land-covered portions of the domain, which renders challenging the use of periodic basis functions and may create artifacts near land-ocean boundaries.

Concurrently, Wang et al. (2024) introduced a transformer-based model tailored for oceanic applications, demonstrating performance that rivals that of leading operational global ocean forecasting systems. Similar advances are being made in data-driven prediction of sea ice cover in the polar oceans (Anderson et al., 2021; see also Bertino et al., this issue). This body of

work signifies the emergence of a promising research avenue in fully data-driven ocean modeling, despite it still lagging considerably behind the advancements seen in weather forecasting. We posit that the drive of fully data-driven solutions in NWP by private sector companies is related to the prospect of high-stakes / high-reward applications. Such applications for ocean predictions should be better articulated to attract similar research efforts. Careful evaluation of skill, such as now being discussed more comprehensively in NWP (e.g., Charlton-Perez et al., 2024) will also be required for operational ocean

prediction.

Another challenge presents the extension of ML applications to seasonal, inter-annual and multi-decadal - i.e. climate - time scales (see e.g., the discussion in Gentine et al., 2021; Beucler et al., 2024; Subel and Zanna, 2024). Here, the increased need of models or invariant operators (physics-based or surrogates) to conserve fundamental properties (mass, energy, momentum, active tracers) puts severe demands on ML approaches. Arguably, as these approaches increasingly incorporate physical

knowledge, they will converge to the realm of classical inverse methods (Willcox et al., 2021).

## 2.4 Benchmarking forecast models

Data-driven forecasting in meteorology - and to some extent in oceanography - is proceeding at a breathtaking pace. The use of different approaches, different training data, and different performance metrics complicates objective assessment of the different works at the present time. Recognizing the need for standardized evaluation has led to the proposition of common

evaluation benchmarks that encompass both data-driven and "traditional" forecasting in weather prediction (Dueben et al., 2022; Rasp et al., 2020, 2024), as well as climate model emulation (Yu et al., 2024). These benchmarks comprise common data sets, open-source evaluation workflows, and common evaluation metrics. Similar benchmarking efforts in ML-driven ocean circulation and surface wave forecasting will be equally important to advance the field and establish standardized evaluation metrics.

## 3 The role of surrogate models in digital twins

The concept of digital twins (DTs) is rapidly gaining traction within the ocean science community and Earth system science more broadly (e.g., Bauer et al., 2021a, 2021b). Because of the differing view of what constitutes a DT in the recent literature, we here adopt and emphasize the definition from NASEM (2022) (see also Niederer et al., 2021; National Academies of Sciences, Engineering, and Medicine, 2023), a DT is "a set of virtual information constructs that mimics the structure, context and behaviour of an individual/unique physical asset, or a group of physical assets, is dynamically updated with data from its physical twin throughout its life cycle and informs decisions that realize value. A digital twin is highly dynamical, mimicking the time evolution of its physical asset (PA) via advanced simulation and emulation capabilities; it is updated by ingesting vast amounts of observational data of diverse types; and it enables WHAT-IF queries and multiple realizations to support prediction of responses of the PA to hypothetical perturbations with quantified uncertainties."

Virtually all aspects of ocean forecasting – and ML opportunities therein – may be viewed through the DT lens, from the need to generate high-fidelity simulations or digital representations, ingesting, i.e., assimilating large, heterogeneous data streams, the development of fast surrogates or emulators to either accelerate simulations or provide comprehensive uncertainty estimates, to the generation of diagnostic data that create value for (possibly rapid) decision support.

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

## Competing interests

The contact author has declared that none of the authors has any competing interests.

### Data and/or code availability

This can also be included at a later stage, so no problem to define it for the first submission.

### Authors contribution

This can also be included at a later stage, so no problem to define it for the first submission.

### Acknowledgements

This can also be included at a later stage, so no problem to define it for the first submission.