# Peer review of "Crafting the Future: Machine Learning for Ocean Forecasting"

_State of the Planet, 2024_

## Referee Comment (RC1)

Review of

"Crafting the Future: Machine Learning for Ocean Forecasting"

By

Patrick Heimbach, Fearghal O'Donncha, Jose Maria Garcia-Valdecasas, Alain Arnaud, Liying Wan

**My summary and recommendation**

The authors present a "state of the art" view of machine learning with regard to how it has been and can be used for ocean prediction. The manuscript is well organized, and has mostly up-to-date citations - which is a serious challenge in this rapidly developing field. I think the manuscript should be eventually published. However, I have two major comments that I believe should be addressed before it can be published.

**Major Comments**

- The list of architectures given in section 2.2 should be revised. On the one hand, considering "blocks" or components of the network, it is not really a comprehensive list since it ignores

    - Graph Neural Networks (i.e., the backbone of GraphCast, one of the leading atmospheric emulators (Lam et al., 2023))
    - Transformers, which have been revolutionary in other ML/AI fields like natural language processing and image recognition/generation, and serves as the backbone for some of the leading atmospheric emulators like Pangu Weather (Bi et al., 2023), FuXi (Chen et al., 2023), FengWu (Chen et al., 2023), and in a sense FourCastNet (although FNOs/AFNOs/SFNOs tend to "feel" different than other transformers; Pathak et al., 2022)
    - Regarding Convolutional Networks, at least some of the various works from Dale Durran's group should be listed, especially since the papers led by Weyn helped kick off the ML weather emulation generally. For example (Weyn et al., 2019; Weyn et al., 2020; Weyn et al., 2021; Karlbauer et al., 2023; Wang et al., 2024).

    The architectures above have proven skill in emulating medium range weather, whereas two of the architectures listed (LSTMs and Reservoir Computing) have not. Given that the authors state that the ocean prediction workflow mirrors that of NWP, I think it is therefore natural to make this comparison to Medium Range Weather. Moreover, for a more generic list like what is shown in this paper one could put LSTMs and Reservoir Computers under the same architecture umbrella, since they are both Recurrent Neural Networks, and therefore share the same inductive biases as outlined by Battaglia et al., 2016. As a final note on the RNNs, if Reservoir Computing is included in this list, then it

may be useful to include references that focus on GFD related emulation rather than just Lorenz-like systems. For example Arcomano et al., 2020 & Smith et al., 2023 might be useful to some readers.

However, note that the distinction between these "architectures" is somewhat soft here, since for example:
- Pangu uses a Transformer as the processor, but uses CNNs during the encoding steps.
- Both the AIFS (Lang et al., 2024) and GenCast (Price et al., 2024) uses a Graph network based encoder/decoder, but uses a Transformer as the processor (where for GenCast this is used as the Diffusion Denoiser rather than an emulator, but the overall idea remains).
- The emulator developed by Karlbauer et al., (2023) uses a U Net (i.e., convolutional) backbone, but also includes Gated Recurrent Unit (GRU) blocks, which is a Recurrent Neural Network "architecture".

This is what I mean when I refer to these as "blocks" of a network architecture - the components can be used interchangeably to make up a distinct emulator architecture. Therefore, it's somewhat unclear when these are listed as distinct components.

On top of that, I do not think that GANs should be given in the same list for several reasons. First of all, this refers to a completely different method of training a network, rather than an architectural "block" as noted above, and so it should be given in a distinctly different list. Secondly, GANs are more or less not being used anymore, in favor of Diffusion models (e.g., GenCast as in Price et al., 2024). That said, I believe the more useful distinction here, which is separate from the list of architectural blocks noted above, is the distinction between emulators that are trained to be deterministic forecast models (which most of these citations fit under), versus generative models, i.e. models that are designed to be probabilistic. A short list of popular generative model frameworks would include: Variational Auto Encoders, Generative Adversarial Networks, and Diffusion Models. Again, this is distinct from the list above because, for example, one could use any of the blocks above as the denoiser in a Diffusion model - UNets, Graph networks, transformers, or some combination thereof could be used.

- This is somewhat subjective, but I strongly oppose the "Hard AI" and "Soft AI" terminology that is used. I recognize that the authors are using this in reference to the work by Chantry et al., (2021). However, I think that these words tend to obfuscate the task at hand, and I would instead suggest the following terms: "Emulators" or "Purely data driven prediction models" or something like that instead of "Hard AI", and "Hybrid Physics and ML" or something like that for "Soft AI". Other than the fact that Hard and Soft do not actually describe the learning task, the main reason that I do not like these terms is because designing a purely data driven model is not exactly harder than developing a hybrid model. In fact, designing a hybrid model should be **much** harder because it requires the dynamics/physics based model to have an adjoint, so that

gradient information can flow between the state vector (which is used to compute the loss function) and the parameter values (i.e., neural network weights). There are very few models which are actually fit for this task, and designing them to be so is extremely challenging. Moreover, hybrid models are bound to the CFL condition, as any dynamical model is, and therefore require much shorter timesteps than purely data driven counterparts.

I don't mean to belabor the point, but I think the numbers really do the talking. Consider the comparison between the purely data driven emulator GraphCast (Lam et al., 2023) and the hybrid physics/ML model NeuralGCM (Kochkov et al., 2024). For training, GraphCast required 4 weeks of training on 32 TPU devices (128 TPU weeks), whereas NeuralGCM required 3 weeks of training on 256 TPUs (768 TPU weeks) for a model resolution that is ~3 times as coarse (0.7 degree vs 0.25 degree) and uses 5 million fewer parameters (31 million vs 36 million). In terms of on-demand cost using Google Cloud Platform, this amounts to a rough estimate of 21,500 USD to train GraphCast, vs 129,000 USD to train NeuralGCM.

All that to say, the "soft" problem actually seems a bit harder to me.

**Minor comments**

- Line 30: "prerequisite to for" -> "prerequisite for"
- Line 71: I would also include the following work in the list of hybrid dynamics/ML models: Arcomano et al., 2023
- Line 78: PDE -> PDEs.
- Line 83: I think the "i.e." should actually be "e.g." since MSE based loss (i.e., L2 norm loss) is only one example. Another popular choice is an L1 norm loss, although this has similar detrimental effects like producing overly blurred output. In generative applications, though, more generic loss functions are being used.
- Line 138: Since the positive side of FNOs is listed, and since this is for an ocean audience, I would also list their main drawback for ocean applications - that they will be challenging (and maybe infeasible) to use in the ocean due to non periodicity and continental boundaries. This can create artifacts at the boundaries, which would limit their stability, and overall attractiveness, in comparison to atmosphere applications.

**References**

Arcomano, T., Szunyogh, I., Pathak, J., Wikner, A., Hunt, B. R., & Ott, E. (2020). A Machine Learning-Based Global Atmospheric Forecast Model. Geophysical Research Letters, 47(9), e2020GL087776. https://doi.org/10.1029/2020GL087776

Arcomano, T., Szunyogh, I., Wikner, A., Hunt, B. R., & Ott, E. (2023). A Hybrid Atmospheric Model Incorporating Machine Learning Can Capture Dynamical Processes Not Captured by Its

Physics-Based Component. Geophysical Research Letters, 50(8), e2022GL102649.
https://doi.org/10.1029/2022GL102649

Bi, K., Xie, L., Zhang, H., Chen, X., Gu, X., & Tian, Q. (2023). Accurate medium-range global weather forecasting with 3D neural networks. Nature, 619(7970), 533–538.
https://doi.org/10.1038/s41586-023-06185-3

Chen, K., Han, T., Gong, J., Bai, L., Ling, F., Luo, J.-J., et al. (2023, April 6). FengWu: Pushing the Skillful Global Medium-range Weather Forecast beyond 10 Days Lead. arXiv.
https://doi.org/10.48550/arXiv.2304.02948

Chen, L., Zhong, X., Zhang, F., Cheng, Y., Xu, Y., Qi, Y., & Li, H. (2023). FuXi: a cascade machine learning forecasting system for 15-day global weather forecast. Npj Climate and Atmospheric Science, 6(1), 1–11. https://doi.org/10.1038/s41612-023-00512-1

Lam, R., Sanchez-Gonzalez, A., Willson, M., Wirnsberger, P., Fortunato, M., Alet, F., et al. (2023). Learning skillful medium-range global weather forecasting. Science, 0(0), eadi2336.
https://doi.org/10.1126/science.adi2336

Lang, S., Alexe, M., Chantry, M., Dramsch, J., Pinault, F., Raoult, B., et al. (2024, June 3). AIFS - ECMWF's data-driven forecasting system. arXiv. Retrieved from
http://arxiv.org/abs/2406.01465

Karlbauer, M., Cresswell-Clay, N., Moreno, R. A., Durran, D. R., Kurth, T., & Butz, M. V. (2023, September 11). Advancing Parsimonious Deep Learning Weather Prediction using the HEALPix Mesh. arXiv. https://doi.org/10.48550/arXiv.2311.06253

Pathak, J., Subramanian, S., Harrington, P., Raja, S., Chattopadhyay, A., Mardani, M., et al. (2022). FourCastNet: A Global Data-driven High-resolution Weather Model using Adaptive Fourier Neural Operators. arXiv:2202.11214 [Physics]. Retrieved from
http://arxiv.org/abs/2202.11214

Price, I., Sanchez-Gonzalez, A., Alet, F., Andersson, T. R., El-Kadi, A., Masters, D., et al. (2024, May 1). GenCast: Diffusion-based ensemble forecasting for medium-range weather. arXiv.
Retrieved from http://arxiv.org/abs/2312.15796

Smith, T. A., Penny, S. G., Platt, J. A., & Chen, T.-C. (2023). Temporal Subsampling Diminishes Small Spatial Scales in Recurrent Neural Network Emulators of Geophysical Turbulence.
Journal of Advances in Modeling Earth Systems, 15(12), e2023MS003792.
https://doi.org/10.1029/2023MS003792

Wang, C., Pritchard, M. S., Brenowitz, N., Cohen, Y., Bonev, B., Kurth, T., et al. (2024, June 12). Coupled Ocean-Atmosphere Dynamics in a Machine Learning Earth System Model. arXiv.
https://doi.org/10.48550/arXiv.2406.08632

Weyn, J. A., Durran, D. R., & Caruana, R. (2019). Can Machines Learn to Predict Weather? Using Deep Learning to Predict Gridded 500-hPa Geopotential Height From Historical Weather Data. Journal of Advances in Modeling Earth Systems, 11(8), 2680–2693. https://doi.org/10.1029/2019MS001705

Weyn, J. A., Durran, D. R., & Caruana, R. (2020). Improving Data-Driven Global Weather Prediction Using Deep Convolutional Neural Networks on a Cubed Sphere. Journal of Advances in Modeling Earth Systems, 12(9), e2020MS002109. https://doi.org/10.1029/2020MS002109

Weyn, J. A., Durran, D. R., Caruana, R., & Cresswell-Clay, N. (2021). Sub-Seasonal Forecasting With a Large Ensemble of Deep-Learning Weather Prediction Models. Journal of Advances in Modeling Earth Systems, 13(7), e2021MS002502. https://doi.org/10.1029/2021MS002502

---

## Referee Comment (RC2)

[referee-annotated manuscript omitted]

---

## Author Comment (AC1)

**Response to Reviewers**

**Reviewer #1**

We thank the reviewer for the careful read of the manuscript.

Before addressing Reviewer #1's comments we note that we have reorganized the manuscript subtantially in order to incorporate Reviewer #1's detailed suggestions. We have also informed the editorial team and wish to state here that we have added Timothy A. Smith, NOAA Physical Sciences Lab, Boulder, CO, USA, to the team of authors.

In the following we address the reviewer's comments (reviewer's comments in red, our response in black).

Major comments:

The list of architectures given in section 2.2 should be revised. On the one hand, considering "blocks" or components of the network, it is not really a comprehensive list since it ignores
- Graph Neural Networks (i.e., the backbone of GraphCast, one of the leading atmospheric emulators (Lam et al., 2023))
- Transformers, which have been revolutionary in other ML/AI fields like natural language processing and image recognition/generation, and serves as the backbone for some of the leading atmospheric emulators like Pangu Weather (Bi et al., 2023), FuXi (Chen et al., 2023), FengWu (Chen et al., 2023), and in a sense FourCastNet (although FNOs/AFNOs/SFNOs tend to "feel" different than other transformers; Pathak et al., 2022)
- Regarding Convolutional Networks, at least some of the various works from Dale Durran's group should be listed, especially since the papers led by Weyn helped kick off the ML weather emulation generally. For example (Weyn et al., 2019; Weyn et al., 2020; Weyn et al., 2021; Karlbauer et al., 2023; Wang et al., 2024).

We thank the reviewer for their detailed suggestions. We have conducted a major restructuring and extension of the manuscript to accommodate all of the reviewer's comments. As a result, we have also extended the list of work cited.

The architectures above have proven skill in emulating medium range weather, whereas two of the architectures listed (LSTMs and Reservoir Computing) have not. Given that the authors state that the ocean prediction workflow mirrors that of NWP, I think it is therefore natural to make this comparison to Medium Range Weather. Moreover, for a more generic list like what is shown in this paper one could put LSTMs and Reservoir Computers under the same architecture umbrella, since they are both Recurrent Neural Networks, and therefore share the same inductive biases as outlined by Battaglia et al., 2016. As a final note on the RNNs, if Reservoir Computing is included in this list, then it may be useful to include references that focus on GFD related emulation rather than just

Lorenz-like systems. For example Arcomano et al., 2020 & Smith et al., 2023 might be useful to some readers.
We concur. In our revised and restructured manuscript, we have incorporated this comment and the related references. We have also accounted for the reviewer's comments (next paragraph of their review) regarding GAN's and the rise of diffusion models in our revised version.

This is somewhat subjective, but I strongly oppose the "Hard AI" and "Soft AI" terminology that is used. [+ rest of paragraph].
Although Chantry et al. (2021) attribute a different meaning to the terms "soft" vs. "hard", we concur with the reviewer that these terms are non-descriptive and can mean different things to different people. Following the reviewer's suggestions, we have made the following replacements:
- "soft AI" to "Hybrid physics-ML models"
- "hard AI" to "Purely data-driven models"

We've also added the following note in the revised manuscript:
> "Chantry et al. (2021) have used the terms "soft AI" versus "hard AI". We avoid the somewhat non-descriptive or ambiguous terminology in order not to give a false sense of which of these approaches is "harder" to realize."

Minor comments:

Line 30: "prerequisite to for" -> "prerequisite for"
Corrected.

Line 71: I would also include the following work in the list of hybrid dynamics/ML models: Arcomano et al., 2023
Added.

Line 78: PDE -> PDEs.
Done.

Line 83: I think the "i.e." should actually be "e.g." since MSE based loss (i.e., L2 norm loss) is only one example. Another popular choice is an L1 norm loss, although this has similar detrimental effects like producing overly blurred output. In generative applications, though, more generic loss functions are being used.
Thank you, we have modified the text to reflect this.

Line 138: Since the positive side of FNOs is listed, and since this is for an ocean audience, I would also list their main drawback for ocean applications - that they will be challenging (and maybe infeasible) to use in the ocean due to non periodicity and continental boundaries. This can create artifacts at the boundaries, which would limit their stability, and overall attractiveness, in comparison to atmosphere applications.

A very good point, which we have adopted in the revised manuscript. We have added the following sentence:

> "A drawback of FNOs applied to ocean (unlike atmospheric) modelling is the existence of land-covered portions of the domain, which renders challenging the use of periodic basis functions and may create artifacts near land-ocean boundaries."

---

## Author Comment (AC2)

**Response to Reviewers**

**Reviewer #2**

We thank the reviewer for the careful read of the manuscript.

Before addressing Reviewer #2's comments we note that we have reorganized the manuscript subtantially in order to incorporate Reviewer #1's detailed suggestions. We have also informed the editorial team and wish to state here that we have added Timothy A. Smith, NOAA Physical Sciences Lab, Boulder, CO, USA, to the team of authors.

In the following we address the reviewer's comments (reviewer's comments in red, our response in black).

This is an interesting review of the current status of Machine Learning for Ocean Forecasting, especially for people from outside the subject domain.
We thank the Reviewer for their comment.

The main concern I have is that level of detail of the discussion is rather uneven. For example, the discussion of Sec. 3 "Enhancing data assimilation with ML algorithms" seems just a placeholder for further development. The alternative is that not a lot of activity has been going in the field, in which case this should be stated.
The impression is mostly correct. Most efforts have been dedicated to surrogate modeling, either of the full ocean GCM or of components – parameterization schemes in particular – of the model. Whereas hybrid DA/ML methods have not been as widespread yet, they are an important application of ML and we disuss them here.
   Following the reviewer's suggestion, we are now stating the relative paucity of related activities in ocean modeling. Among others, the revised manuscript now contains the following statement:
   "The use of hybrid DA/ML approaches, be it in the context of ensemble DA or adjoint-based methods (e.g., 4DVar) presents substantial algorithmic hurdles (e.g., availability of a differentiable dynamical core in the context of adjoint-based DA), which explains the relative paucity of such studies to date compared to purely data-driven methods.

Other comments are posted in the attached annotated version of the manuscript.
We have addressed all comments in the annotated PDF. Below, we take up those comments in need of a response.

Line 86:
Maybe some high level explanation of how this approach "tries to solve differential equations using NNs" would useful here.
We have now removed this generic statement in favor of a more detailed list of ML approaches that have been explored in the context of numerical weather prediction.

Line 115:

That is a very fashionable definition! In practice DTs are effectively high resolution versions of standard NWP or Earth System numerical models and their ensemble implementation. So I would take this definition off, as it does not bring additional information to the discussion.

We disagree with this statement, in particular with the notion that DTs "are effectively high-resolution versions of standard NWP or Earth System numerical models and their ensemble implementation". The US National Academies' (2023) which we are citing makes the specific point that this is a misguided concept of DTs, although it is indeed used by various groups. We choose to stick to the definition (and vision) of DTs as laid out in the National Academies' (2023) report.

Line 137:

Would be interesting to provide examples of where this ability to incorporate physical laws and constraints has been demonstrated.

In the revised and restructured version, we now clarify how ML approaches are combined with physical laws in new section 2.1 "Hybrid physics-ML models: enhancing forecast models and data assimilation with ML algorithms".